# Trajectories of Competitive Employment of Autistic Adults through Late Midlife

**DOI:** 10.3390/healthcare12020265

**Published:** 2024-01-20

**Authors:** Emily J. Hickey, Leann Smith DaWalt, Jinkuk Hong, Julie Lounds Taylor, Marsha R. Mailick

**Affiliations:** 1University Center for Excellence in Developmental Disabilities, Waisman Center, University of Wisconsin-Madison, 1500 Highland Ave., Madison, WI 53705, USA; lesmith2@wisc.edu; 2Waisman Center, University of Wisconsin-Madison, 1500 Highland Ave., Madison, WI 53705, USA; jhong@waisman.wisc.edu (J.H.); marsha.mailick@wisc.edu (M.R.M.); 3Pediatrics and Psychiatry & Behavioral Science, Vanderbilt University Medical Center, 1211 Medical Center Dr., Nashville, TN 37232, USA; julie.l.taylor@vumc.org

**Keywords:** autism, competitive employment, midlife, adulthood, accelerated longitudinal design

## Abstract

Autistic adults experience challenges in maintaining employment; however, little is known about patterns of competitive employment through late midlife. This longitudinal study examined the change in hours of competitive employment for a cohort of autistic adults over a 22-year period. The study’s aims were to provide a fine-grained analysis of competitive employment patterns, to determine whether there was age-related change, and to test whether trajectories differed between those with and without intellectual disability (ID). Using an accelerated longitudinal design, trajectories of hours of competitive employment were estimated from young adulthood through late midlife in a community-based cohort (*n* = 341; 1327 observations). Results indicated a significant curvilinear trajectory of age-related change in hours of competitive employment, with differences between those with and without ID. For those without ID, the number of competitive employment hours increased from young adulthood until early midlife, then leveled off and decreased into late midlife. For those with ID, engagement in competitive employment was low throughout. Although competitive employment is just one option for vocational engagement, it is a goal often articulated by autistic adults who seek entry into the general workforce. The present research reveals their degree of engagement in the competitive workforce across the decades of adulthood.

## 1. Introduction

With over 5.5 million autistic adults in the U.S. [1] and the rising prevalence of autism [2], there is an increasing focus on adulthood within this population. Many autistic adults experience poor outcomes in vocational activities [3,4], social participation [5,6], and independent living [7,8] that, for some, continue and worsen throughout adulthood [7,9,10,11]. Many of these poor outcomes are related to disengagement from vocational activities [7,12]. Competitive employment is often difficult for autistic adults to obtain [7,12,13] and is rarely maintained over time [9,14,15]. In this paper, we focus on competitive employment, defined as paid work without support. Past research has identified independence in employment as a self-reported goal for many autistic youth [16]. Additionally, competitive employment reflects the goal of many supported employment agencies, namely, to phase out job support over time and transition into the open labor market [17].

In the general population, employment predicts success in other areas, including financial autonomy, independent living, and social connectedness [18,19]. Unemployment, on the other hand, is associated with increased mental health concerns [20]. Within autism populations, competitive employment has also been linked to well-being [3], providing a social outlet in adulthood, strengthening skills and resources necessary for independent living [4], and increasing feelings of autonomy [21].

We focus here specifically on competitive employment, recognizing that there are other options for vocational engagement for autistic adults. For the present study, we seek to quantify engagement in the general workforce at various stages of adulthood. We further focus the present study on longitudinal patterns of engagement in competitive employment across the decades of adulthood. A growing number of studies have examined vocational and employment outcomes over time [3,9,10,22]; however, to our knowledge, only one extended into mid-to-older age [22] and did not find an age effect on engagement in employment by autistic adults. Low rates of employment were evident across the 20-year study, with only 28% of the sample reporting that they ever worked. Notably, this sample included only adults with an average or above IQ (70+) when first diagnosed and did not consider hours of employment. Clarke and colleagues (2021) followed a cohort of autistic adults from ages 18–28 and similarly found no effect of age on vocational trajectories based on the level of independence of the type of work as assessed by the Vocational Index [13]. Both studies included all forms of work, including volunteer work, agency-based work, and supported employment in addition to competitive employment.

The current study extends past research by including participants as young as 18 to as old as 68 years. With midlife often involving a peak in employment position and earnings within the general population [23], it is important to consider midlife when measuring age effects on employment in autistic populations as well. Midlife is a period that is often characterized in the general population as between 40 and 60 years of age, plus or minus 10 years [24]. It is seen as a “pivotal period in the life course in terms of balancing growth and decline, linking earlier and later periods of life [23]”. To our knowledge, no previous study has examined trajectories of competitive employment in autistic adults with and without ID through midlife.

### The Current Study

The goal of the current study was to identify and describe the change in hours of competitive employment, defined here as paid employment in the community without support, of autistic individuals from young adulthood into late midlife. In addition to providing descriptive data, we used an accelerated longitudinal design approach with a cohort followed prospectively over a 22-year period to characterize the patterns of age-related change in hours of competitive employment. We investigated whether there was linear or curvilinear age-related change and evaluated whether trajectories differed between those with and without ID. Through a series of follow-up sensitivity analyses, we explored: (1) whether the slope of the trajectories differed depending on the age of the participant when the study began, with the goal of identifying possible cohort effects; (2) whether the pattern of results was similar using only cases with complete data; and (3) whether the pattern of results was similar without the final wave of data, which was collected during the peak of the COVID-19 pandemic (2021–2022).

## 2. Methods

### 2.1. Data and Sample

The current study used eight waves of data from an ongoing longitudinal study of 406 autistic adolescents and adults, who ranged in age from 10 to 52 at the start of the study, and their families [25,26]. All participating families initially lived in Massachusetts and Wisconsin and had a son or daughter age 10 or older with an autism diagnosis given by an educational or health professional, which was confirmed with a researcher-administered Autism Diagnostic Interview-Revised (ADI-R) [27] profile consistent with the diagnosis. The study began in 1998, and data collection has thus far extended over 22 years. When the study began, 94.6% met the criteria for a diagnosis of autistic disorder. The remaining 5.4% were determined to have ADI-R profiles consistent with a diagnosis of Asperger’s disorder or Pervasive Developmental Disorder Not Otherwise Specified (PDD-NOS) [28].

Thus far, the larger study has completed nine repeated waves of data collection. Eight of the waves from the larger study provided data relevant to the current study (data about hours of competitive employment were not included in one wave). The eight waves of data for the current study (here referred to as Time 1 through Time 8) spanned 22 years. The present analysis used repeated measures that were collected over those eight study waves with mothers (96%) or another family member, including in-home interviews and self-administered questionnaires, providing unique, rich descriptive data on the competitive employment of the autistic adults in the sample after high school exit (*n* = 341, 83.9% of the full sample, number of observations = 1327). This study differs from previous reports from the larger study by including an additional 8 years of data [4,9,10,15,25]. Thus, we are able to investigate not only entry into the workforce but also competitive employment status well into midlife. The present study additionally builds on our past research by distinguishing competitive employment from overall vocational engagement (which we previously measured by the Vocational Index) [13], which included volunteer work, agency-based work, and supported employment, in addition to competitive employment [9,10,12,13].

The youngest members of the full sample were still in high school when the study began; most of these cases were eventually included in the present analysis, starting with the wave of data collection after they exited high school. High school exit was defined as no longer receiving school-based services or participating in classes or programming at the high school level, as determined by the mother’s report. See Table 1 for the sample sizes at each time point. For example, data from a participant who was age 17 and still in high school at Time 1 would not have been analyzed at that first wave; however, the same participant’s data from Time 2 would be included if they had exited high school at that second wave of data collection, and their data would continue to be included from Time 2 onward. Table 1 also shows the timing of each wave of data collection analyzed for the present study, along with the average ages and age ranges of the autistic individuals included in each wave. As described below, we conducted a sensitivity analysis including only participants for whom there was complete data (i.e., they contributed data at all time points after they exited high school; *n* = 170) and found similar patterns of results.

The majority of the autistic adults in the current sample were males (72.9%), and almost three-fourths (71.9%) had a co-occurring diagnosis of ID. Nearly two-thirds (65.0%) lived with their mothers (and often other family members) at Time 1. The majority of the families were White non-Hispanic (92.6%). There was significant socioeconomic heterogeneity, with the median annual household income at Time 1 between USD 50,000 and USD 60,000. Notably, 11.5% earned less than USD 20,000 per year when the US poverty line was USD 17,050 for a family of four [29].

### 2.2. Measures

#### 2.2.1. Dependent Variable

Hours of engagement in competitive employment. The main outcome was hours of competitive employment. Mothers reported on the number of hours their son or daughter engaged in competitive work in the community. The number of hours per week was initially categorized as 0, 1–10, 11–20, 21–30, and 31+. Consistent with the approach used in other studies, these hours were recoded to the midpoint of each of the categories to aid in analysis and interpretation (0, 5, 15, 25, 35 h) [30,31,32].

#### 2.2.2. Predictor Variables

Time-varying age. The main predictor was age at each wave of data collection. The age of each individual was calculated from the date of birth to the date of that individual’s data collection at each time point.

ID status. ID status was coded as 0 = no intellectual disability, 1 = intellectual disability, and was determined using standard scores of 70 or below on the Wide Range Intelligence Test (WRIT) [33] and the Vineland Screener [34], consistent with diagnostic guidelines [35]. For individuals with scores greater than 70 on either measure or when either of the measures for the individual was missing, clinical consensus among three PhD-level psychologists was reached to determine ID status based on a review of medical and educational records.

#### 2.2.3. Covariates

Time 1 Age. In addition to time-varying age at each time point being used as a predictor, age at Time 1 (constant) was included as a covariate, regardless of when their data began to be included in trajectory analysis (i.e., after high school exit).

Sex. Sex was coded as 0 = male, 1 = female.

#### 2.2.4. Sensitivity Analysis

Time 1 Age. Age at Time 1 was also used in sensitivity analyses to evaluate whether the trajectories differed depending on the age of the participant when the study began. To do so, we tested the effect of a three-way interaction: Time 1 age X age at each wave of data collection × ID status. This provided insight into possible cohort effects in the age-related trajectories (for example, whether individuals who were young adults at the start of the study showed a different pattern of change as compared to those who were older when the study began).

### 2.3. Data Analysis

#### 2.3.1. Descriptive Data

Our first goal was to provide rich descriptive data about engagement in competitive employment as autistic adults move through adulthood and midlife. We calculated the means and standard deviations of hours of competitive work by ID status across time in the full analytic sample. We then classified participants into two groups—those who ever had a competitive job and those who never held such a job at any time during the study period.

#### 2.3.2. Trajectories of Age-Related Change in Hours of Engagement in Competitive Employment

Our second goal was to estimate the trajectory of hours of competitive employment for autistic adults with and without ID using an accelerated longitudinal design (ALD), also referred to as a cohort-sequential design or cross-sequential design. Because ALD estimates a single long-term longitudinal trajectory by combining the trajectories of each individual covering different ages, it estimates a full combined growth trajectory [36]. The current analysis included individuals as young as 18 at Time 1 and as old as 68 at Time 8; thus, we are able to estimate age-related trajectories spanning 50 years. In the current sample, there were no statistically significant differences in competitive employment based on sex. Mixed-effects growth curve models with polynomial functions of age were estimated to assess linear, quadratic, and cubic trajectories. The predictor variable, covarying age across time (linear, quadratic, and cubic terms), was specified as random. Person-level fixed variables (age at Time 1, sex, and ID status) were added to predict the baseline (intercept) differences between measures and to control for their effects on competitive employment. Cross-level interaction terms between age and ID status were then separately added to the models to evaluate the independent effects of ID status on these trajectories. Models 1–3 estimated the linear, quadratic (with a term of age-squared added), and cubic (with a term of age-cubed added) age effects, respectively, to assess whether age was best estimated as a linear or curvilinear function. Models 4–6 included the addition of age × ID interaction terms to assess whether the age-related linear or curvilinear trajectories differed by ID status. The best-fitting model was selected based on Akaike Information Criteria (AIC) and Bayesian Information Criteria (BIC) values, as well as parsimony, following the approach of Joiner, Bergeman, and Wang [37]. We report the results of the best model.

In the main ALD analysis, all 341 individuals in the study sample were included, regardless of the number of data points they contributed post-high school exit. Those with just one data point contributed to the estimation of the intercepts and the effect of age at the start of the study on the intercept, but they did not contribute to estimates of the trajectories. The greater the number of data points, the more influential the case was to the estimation of the age-related trajectories. The overall trajectory generated by the equation in the best-fitting model is illustrated graphically.

### 2.4. Sensitivity Analyses

Following these main ALD analyses, we conducted an exploratory sensitivity analysis to probe for potential cohort effects (i.e., to determine if trajectories differed depending on a person’s age when the study began). Time 1 age was added to the age × ID interaction term to create and test a three-way interaction. If this interaction term was significant, it would indicate cohort effects were present. Two additional sensitivity analyses were also performed: one with only participants for whom there was complete data (i.e., they contributed data at all time points after they exited high school; *n* = 170) and one excluding wave 8 data (collected from 2021 to 2022) to test for potential COVID-19 effects. All analyses were conducted using Stata version 17.0 [38].

## 3. Results

### 3.1. Descriptive Findings

As shown in Table 1, the first wave of data collection occurred between 1998 and 2000. The final wave of data collection occurred between 2021 and 2022. The average age of the autistic individuals at Time 1 was 30.4 (SD = 8.4), ranging from 18 to 52. At the final wave of data collection, Time 8, the average age of the autistic individuals was 41.9 (SD = 8.7), ranging from 31 to 68. During the study, 28 autistic individuals died. Data for these individuals were included in the analysis of the waves prior to death. Across waves, adults with ID were significantly older than those without ID (*t*’s ranging from −3.83 to −2.23; *p*’s ranging from <0.001 to 0.027).

Table 2 displays the percentage of participants (with and without ID) who engaged in competitive employment at each wave of data collection. At Time 1, an average of 12.3% of the full sample were competitively employed, with just over one-third of those who did not have ID competitively employed versus 6.5% of those who had ID. By Time 8, two-thirds of those who did not have ID were competitively employed, versus 8.4% of those who had ID. Regardless of ID status, just under 30% held a competitive job at least once during the study period, as shown in the last row of Table 2. However, for adults who did not have ID, approximately 68% held a competitive job at some point during the span of the study period, versus 13.9% of those with ID.

Table 3 presents the hours of competitive employment by ID status across the waves of the study. The averages in Table 3 include 0 h for those who were not competitively employed during a given study wave. The last row of Table 3 indicates the average number of hours of competitive employment across all waves of the study period. Adults without ID averaged just over 9 h of competitive employment per week, and those with ID averaged 1.4 h of competitive employment per week. When data from those who were never employed were removed, those without ID averaged 21 h per week across all waves of the study, whereas those who had ID averaged 17 h per week. Thus, autistic adults with and without ID differed more in their likelihood of competitive employment than in the hours they worked once a job was obtained. Figure 1 and Figure 2 display these data graphically. Figure 1 displays the hours of competitive work by age category and ID status for the full sample, including those who were never competitively employed (i.e., had 0 h of competitive employment). Figure 2 similarly displays hours of competitive work by age category and ID status, but only includes those autistic adults who worked competitively for at least one wave of data collection during the study period.

### 3.2. Trajectories of Age-Related Change in Hours of Engagement in Competitive Employment

Trajectories of hours of competitive employment were analyzed using the ALD approach. Models 1–3 estimated the linear, quadratic, and cubic age effects, and Models 4–6 also included age × ID interaction terms. Model 5, the quadratic model that included the age × ID interaction term, was the best-fitting model in the prediction of age-related trajectories of hours spent in competitive work (AIC: 9810, BIC: 9873).

Table 4 presents the growth curve data for Model 5, indicating that the associations between age and hours spent in competitive work were quadratic and differed for those with and without ID (age × ID coefficient = 0.015, *p* = 0.001). As descriptively illustrated in Figure 3, those without ID showed a curvilinear pattern of hours of competitive work (age coefficient = −0.016, *p* < 0.001), peaking at just over 10 h per week around the mid-30s and then decreasing again. Those with ID did not change their engagement in competitive work across the life course (age coefficient = −0.000, *p* = 0.840), which remained low from adolescence into adulthood and late midlife.

### 3.3. Sensitivity Analyses

#### 3.3.1. Cohort Effects

For our first sensitivity analysis, we explored whether the slope of the trajectories of hours of competitive employment differed depending on the age of the participant when the study began. The three-way interaction term (Time 1 age X age at each wave X ID status) was not statistically significant (coefficient = 0.001, *p* = 0.258), indicating no cohort effects were present.

#### 3.3.2. Complete Cases Only

We next tested whether the pattern of results was similar using only complete cases (*n* = 170) for whom data were available at all waves after they had exited high school. In this subsample, the associations of age and hours of competitive work were consistent with the findings of the full sample analysis (i.e., quadratic and differed for those with and without ID; age × ID coefficient = 0.017, *p* = 0.003). Those without ID showed a curvilinear pattern in hours of competitive work across adulthood (age coefficient = −0.016, *p* = 0.001), and those with ID did not change in their engagement in competitive work across adulthood (age coefficient = 0.000, *p* = 0.972). Based on these results, we concluded that including only data from individuals who provided complete data yielded the same results as an analysis of all available data.

#### 3.3.3. Excluding Time 8

We also tested whether the pattern of results was similar without the final wave of data, as this wave was collected during the peak of the COVID-19 pandemic and could have impacted results. Using the seven waves of data collected prior to COVID-19, the pattern of findings was similar to the main model; however, the quadratic age × ID coefficient was at a trend level (.009, *p* = 0.095). Similarly, those without ID showed a curvilinear pattern of hours of competitive work across the life course, although at a trend level (age coefficient = −0.009, *p* = 0.065), and those with ID did not change in their engagement in competitive work across the life course (age coefficient = 0.000, *p* = 0.923). Thus, we concluded that although the results that did not include the final wave of data were at a trend level, the pattern of results was similar to that observed with this wave of data included.

## 4. Discussion

To the best of our knowledge, the present study is the first to examine trajectories of competitive employment in autistic adults with and without ID through midlife. The results indicated that a large percentage of autistic individuals (70.4%) never engaged in competitive employment during the study period. For those without ID, the number of competitive employment hours increased from young adulthood until early midlife, then leveled off and decreased into late midlife. For those with ID, engagement in competitive employment was low throughout adulthood. By describing the change in hours of competitive employment across the years of adulthood, we are able to provide a complete picture of engagement in the general workforce, not just during young adulthood, as has been the focus of the majority of prior studies [3,9,10], but also into midlife. The current study provides detailed descriptive data and statistically estimated age trajectories of competitive employment in a cohort followed prospectively over a 22-year period, including adults ages 18–68. Thus, the present study was able to identify not only when autistic individuals peak in their hours of engagement with competitive work but also when workforce participation declines, particularly for those without ID.

The current study also extended prior work by contrasting those with and without ID. Those with ID had low rates of competitive work throughout the study, with only 13.9% ever employed competitively as compared with 68.4% of those without ID. Importantly, those with and without ID who did hold a competitive job clocked about the same amount of time at work per week (averaging 17 versus 21 h, respectively). While individuals vary from these averages, the patterns observed in the present study could be helpful to autistic adults, their families, and service providers in setting expectations during the transition into adulthood, in addition to anticipating when competitive work hours may begin to decline. Of note, this study involved both individuals with and without ID from two states with relatively low unemployment rates and strong disability social services; thus, the present findings may present a more positive situation for autistic individuals than in other states [39,40,41].

It is beyond the scope of the current research to suggest why there is low and variable competitive employment among autistic adults. However, it is likely that there are personal, policy, and job market factors impacting rates and patterns over time. Personal factors may include the willingness and ability to work without support [42]. It is also possible that autistic adults make decisions about work based on perceptions of SSI policies and a desire to maintain eligibility for their services [42,43,44]. Job market factors may include the types of accessible jobs available and how many hours employers are willing to offer those jobs.

This study is not without limitations. Although the ALD approach allowed for the inclusion of the entire sample even if they did not participate in each study wave, and nearly half of the sample was retained over the full two-decade study, attrition is a limitation. Generalizations of the present results must also be considered. For example, there were fewer data points reflecting autistic individuals in later midlife than in early midlife. Future studies should extend later into older midlife to better document the trajectories and experiences of autistic adults as they exit the workforce. The lack of ethnic and racial diversity is an additional limitation. However, the current sample was diverse in socioeconomic status, particularly in the inclusion of families living below the poverty line. It is important to note that the current study first collected data in 1998–1999; thus, generalization to samples of autistic adults diagnosed more recently may not be appropriate. Definitions of autism, as well as diagnostic practices, have changed since the adults in our study were diagnosed. It is likely that the range of autistic individuals being diagnosed today differs from those who were diagnosed then. Relatedly, the sample was 73% male, and therefore, the results may not generalize to the experiences of women in competitive employment. Future studies specifically focusing on women and/or larger samples of women are warranted, as they are often underrepresented in autism research. Data included in this research relied on parent reports. Replication with other data sources should be considered in future efforts, including reports directly from autistic individuals. On the other hand, the current research includes autistic adults with a wide range of abilities and functional skills, including those with and without ID. Using parent reports allowed for the inclusion of individuals who were not able to self-report, for whom we know very little about competitive employment, and who are often left out of employment research—and indeed all research—on autistic populations [45].

The current study focused solely on competitive employment in the general workforce. Many autistic adults participate in supported work in the community, and an even larger portion participate in agency-based work [9,12]. Often, they participate in both, with part of the week spent in an agency-based setting, augmented by additional time in supported community employment. Future studies should evaluate supported employment, volunteer work, and agency-based work in order to compare the age trajectories of the number of hours of each. Research on supported employment over time, in particular, will require detailed data and potentially qualitative work on the type, extent, and level of support to understand important variations in supported work. As the goal of the present study was to quantify the hours of paid employment in the general workforce, analysis of other forms of vocational activity was beyond the scope of this research. Balanced with these limitations are several strengths of the present study, including its long longitudinal study period, the prospective approach to data collection, and the inclusion of a community-based sample that was heterogeneous in age, ID status, and socioeconomic status.

## 5. Conclusions

In conclusion, this research revealed the extent of engagement in the competitive workforce among autistic adults with and without ID. Given the impact of unemployment on the health and mental health of adults in the general population, it is critical for new strategies to be developed that facilitate pathways to employment and other meaningful daytime activities for autistic adults. Research to understand pathways to competitive employment is warranted, particularly considering the lack of adequate workplace support that many autistic workers experience [46]. It will be particularly important to investigate skills and experiences that can be facilitated earlier in development to prepare autistic adults for employment in adulthood, such as working on daily living skills or participating in paid employment while in high school, and to determine the extent to which these skills and experiences may predict increased vocational engagement post-high school for autistic individuals with and without ID.

The current study documented clear patterns of employment over time, and these patterns indicate that competitive employment may not be the way to financial stability for many autistic adults. The preferences of autistic adults, families, and employers likely play a large role in these patterns and should be a focus of future investigation. Further, even among those who have worked competitively at some point, few were employed consistently throughout the study period, indicating again that there are unknown factors that are contributing to low and variable competitive employment hours within this population. A close analysis of workplace and community factors that foster employment might point to pathways that lead to consistent and preferred employment. Further, autistic adults across the range of the autism spectrum may benefit from financial policies that provide consistent, comprehensive financial support while also allowing flexibility in the level of employment over time, particularly considering the heterogeneity of autism and the possibility that working full-time might not be feasible or preferable for everyone [47].

The current findings provide autistic individuals, their family members, job coaches, and vocational supporters with comprehensive data on which to base expectations for the reality of what work may look like for many autistic adults. Results show a pattern of a very low number of competitive hours for autistic adults with ID across time and do not show a sustained upward career trajectory in autistic adults without ID. Further, there seems to be a great deal of variation within and across individuals, as documented by the frequent number of “zero” hours, even among those who have worked competitively (see Figure 2). Preparation programs and transition teachers might benefit from keeping these results in mind. Entry into the workforce is not a “once and done” experience but rather an ongoing process of changes in and out of the competitive labor market over time. If having a stable, competitive job is the goal, it may not be possible for the position to be full-time. Managing expectations based on abilities, strengths, preferences, and the availability of inclusive employment options will likely be necessary, in addition to considering a range of daytime activities when planning for the future.

## Figures and Tables

**Figure 1 healthcare-12-00265-f001:**
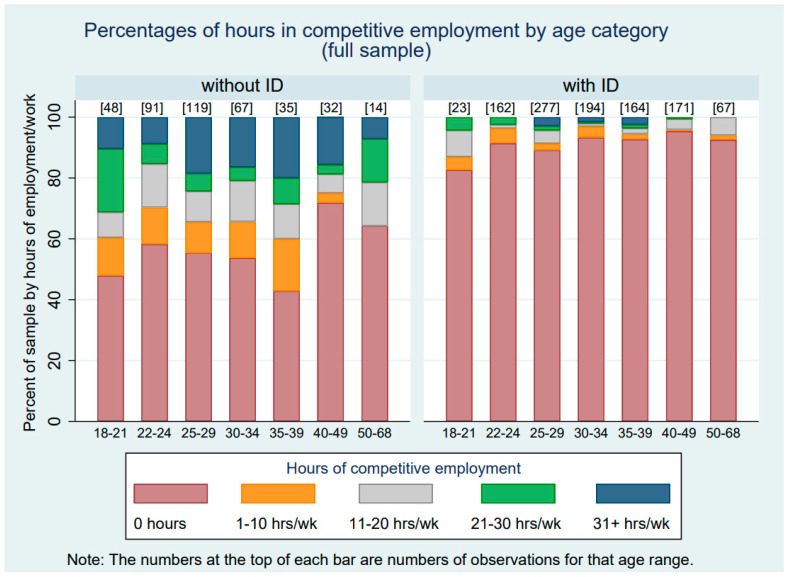
Hours of competitive work by age category and ID status for the full sample.

**Figure 2 healthcare-12-00265-f002:**
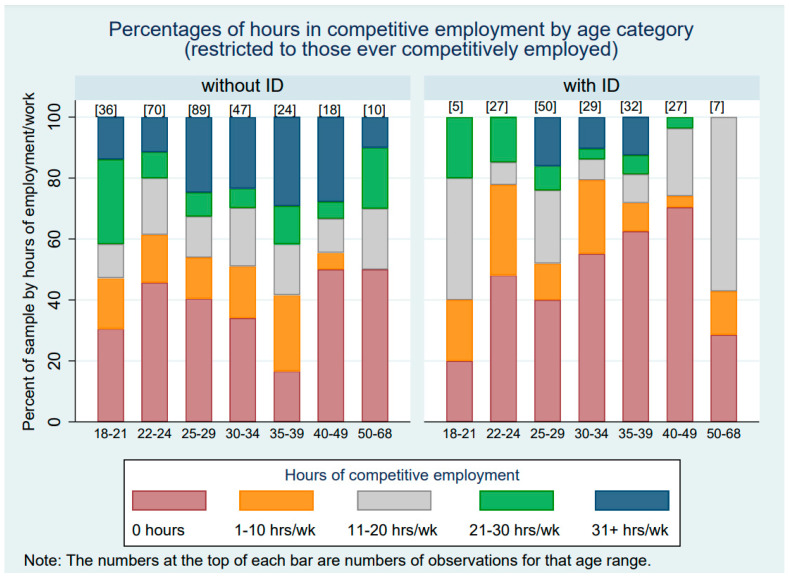
Hours of competitive work by age category and ID status for those who ever worked.

**Figure 3 healthcare-12-00265-f003:**
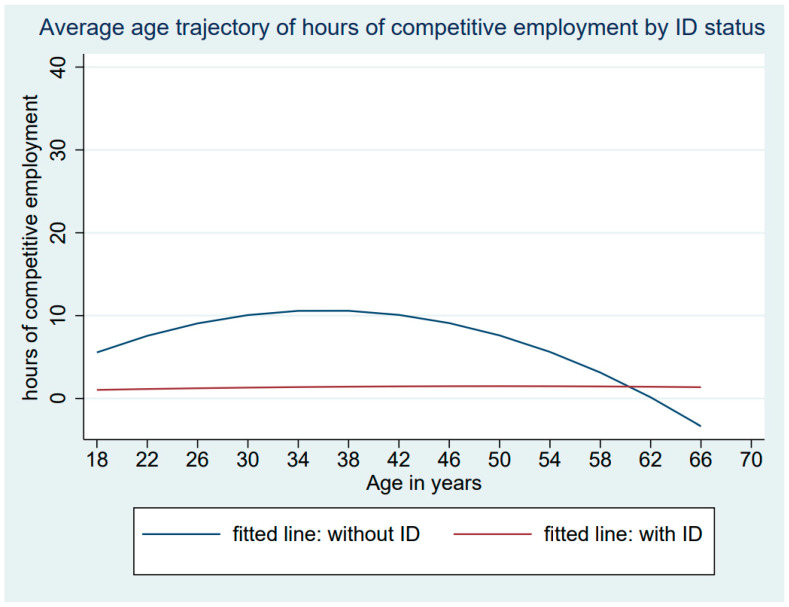
Individual and average trajectories of change in hours of competitive work by ID status.

**Table 1 healthcare-12-00265-t001:** Data collection waves, dates of data collection, and mean ages of autistic adults across time.

Waves of Data Collection (*n*)	Dates of Data Collection	Age of Autistic Adults ^a^		
		Full Sample	ID	No ID
Time 1 (165)	1998–2000	30.4 (8.4) [18,52]	31.1 (8.2) [19,52] (n = 132)	27.6 (8.7) [18,47] (n = 33)
Time 2 (173)	2000–2002	30.6 (8.9) [18,53]	31.6 (8.6) [18,53] (n = 136)	27.1 (9.1) [18,49] (n = 37)
Time 3 (191)	2002–2003	29.9 (9.2) [18,52]	31.6 (8.8) [18,52] (n = 141)	26.3 (8.3) [18,50] (n = 50)
Time 4 (230)	2007–2009	30.1 (8.9) [19,57]	31.3 (9.2) [20,57] (n = 163)	27.0 (7.3) [19,56] (n = 67)
Time 5 (226)	2009–2010	31.0 (8.8) [20,58]	32.2 (9.1) [22,58] (n = 157)	28.1 (7.5) [20,58] (n = 69)
Time 6 (212)	2011–2012	31.8 (8.0) [21,60]	33.1 (8.4) [21,60] (n = 152)	28.5 (5.9) [22,52] (n = 60)
Time 7 (193)	2013–2014	33.7 (8.4) [23,60]	35.1 (8.8) [23,60] (n = 133)	30.6 (6.8) [24,56] (n = 60)
Time 8 (146)	2021–2022	41.9 (8.7) [31,68]	43.1 (8.8) [31,68] (n = 104)	38.8 (7.7) [32,63] (n = 42)

^a^ Means, standard deviations (in parentheses), and ranges (in brackets) are presented.

**Table 2 healthcare-12-00265-t002:** Competitive employment status across waves of data collection.

	Competitive Employment Status ^a^		
Waves of Data Collection	Full Sample	ID	No ID
Time 1	12.3% (*n* = 19) Working 87.7% (*n* = 136) Not Working	6.5% (*n* = 8) Working 93.5% (*n* = 115) Not Working	34.4% (*n* = 11) Working 65.6% (*n* = 21) Not Working
Time 2	12.8% (*n* = 22) Working 87.2% (*n* = 150) Not Working	5.1% (*n* = 7) Working 94.1% (*n* = 128) Not Working	40.5% (*n* = 15) Working 59.5% (*n* = 22) Not Working
Time 3	13.2% (*n* = 25) Working 86.8% (*n* = 164) Not Working	6.4% (*n* = 9) Working 93.6% (*n* = 131) Not Working	32.7% (*n* = 16) Working 67.3% (*n* = 33) Not Working
Time 4	21.8% (*n* = 47) Working 78.2% (*n* = 169) Not Working	12.0% (*n* = 18) Working 88.0% (*n* = 132) Not Working	43.9% (*n* = 29) Working 56.1% (*n* = 37) Not Working
Time 5	23.2% (*n* = 45) Working 76.8% (*n* = 149) Not Working	9.0% (*n* = 12) Working 91.0% (*n* = 121) Not Working	54.1% (*n* = 33) Working 45.9% (*n* = 28) Not Working
Time 6	16.3% (*n* = 34) Working 83.7% (*n* = 175) Not Working	8.1% (*n* = 12) Working 91.9% (*n* = 137) Not Working	36.7% (*n* = 22) Working 63.3% (*n* = 38) Not Working
Time 7	21.4% (*n* = 41) Working 78.6% (*n* = 151) Not Working	9.0% (*n* = 12) Working 91.0% (*n* = 121) Not Working	49.2% (*n* = 29) Working 50.8% (*n* = 30) Not Working
Time 8	24.8% (*n* = 34) Working 75.2% (*n* = 103) Not Working	8.4% (*n* = 8) Working 91.6% (*n* = 87) Not Working	61.9% (*n* = 26) Working 38.1% (*n* = 13) Not Working
Competitive Employment at Any Wave of Data Collection	29.6% (*n* = 101) Ever Worked 70.4% (*n* = 240) Never Worked	13.9% (*n* = 33) Ever Worked 86.1% (*n* = 205) Never Worked	68.4% (*n* = 65) Ever Worked 31.6% (*n* = 30) Never Worked

^a^ Percentages and sample size (in parentheses) are presented.

**Table 3 healthcare-12-00265-t003:** Hours of competitive employment across waves of data collection.

Waves of Data Collection	Hours of Competitive Employment per Week ^a^		
	Full Sample	ID	No ID
Time 1	2.9 (8.4)	1.2 (4.8)	9.5 (14.1)
Time 2	2.7 (8.3)	0.85 (4.5)	9.3 (14.1)
Time 3	2.2 (7.0)	1.1 (5.1)	5.3 (10.0)
Time 4	4.0 (9.1)	1.9 (6.0)	8.7 (12.7)
Time 5	4.5 (9.8)	1.4 (5.5)	11.2 (13.1)
Time 6	3.3 (9.0)	1.5 (5.9)	8.0 (12.8)
Time 7	4.0 (9.4)	1.6 (6.3)	9.4 (12.7)
Time 8	5.2 (10.9)	1.5 (5.8)	13.6 (14.6)
Hours of Competitive Employment Averaged Across All Waves	3.5 (9.0)	1.4 (5.5)	9.3 (13.0)

^a^ Means and standard deviations (in parentheses) are presented.

**Table 4 healthcare-12-00265-t004:** Best-fitting growth curve model for hours of competitive work—Model 5.

	Hours of Competitive Work
Fixed Effects	
T1 age	−0.082 (0.046)
Sex (Female = 1)	−0.974 (0.858)
ID (ID = 1)	0.725 (2.618)
Age	0.813 (0.183) ***
Age-squared	−0.016 (0.004) ***
Age × ID	−777 (0.220) ***
Age-squared × ID	0.015 (0.005) **
Constant	1.378 (2.131)
Random Effects ^a^	
Var. (Age)	0.054 [0.020, 0.149]
Var. (intercept)	58.386 [38.834, 87.783]
Cov. (Age, intercept)	−1.094 [−2.103, −0.084]
Var. (Residual)	29.639 [26.660, 32.951]

** *p* < 0.01; *** *p* < 0.001; ^a^ Estimated variances (Var.) and covariances (Cov.) of random parts of the mixed models are reported with 95% confidence intervals in brackets. Note: ID = Intellectual Disability.

## Data Availability

The datasets presented in this article are not readily availble because they are part of an ongoing study. Requests to access the datasets should be directed to the authors.

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
