# Peer review of "Trajectories of Competitive Employment of Autistic Adults through Late Midlife"

_healthcare, 2024, doi:10.3390/healthcare12020265_

Round 1

Reviewer 1 Report

Comments and Suggestions for Authors

Thank you for the opportunity to review the manuscript titled Trajectories of Competitive Employment of Autistic Adults Through Late Midlife.  The manuscript addresses a pressing issue, competitive employment, within the autistic community and provides valuable data that can assist future researchers, policymakers, and families/individuals in understanding employment patterns in autism through midlife and planning for adult lives.  Below are a few comments/suggestions for author consideration.

·      Table 2 showed a gradual increase in %age of full sample engaged in competitive employment through time 5 followed by a decrease in %age at time 6.  Were the authors able to view data and determine possible reasons for this dip or was it random? The dip was in both ID and No ID.  

·      Based on the data outcomes, particularly for autistic individuals with ID, do the authors have recommendations/thoughts on the practical implications for the population as well as schools/vocational rehab and other agencies in their preparation programs.  It would be interesting to see how many of the individuals in your study had obtained integrated community jobs during their transition years in school (age 14-21) and whether having a community job while in high school predicted increased competitive employment engagement.

·      It appeared that regardless of ID status, autistic individuals are underemployed in terms of hours.   The assumption would be that the pay for the sample would not be enough to sustain them in independent living situations.  This may be a viable recommendation for future research in this area along with ideas for intervention research that can change the trajectories.

Thank you.

Author Response

Reviewer 1

Thank you for the opportunity to review the manuscript titled Trajectories of Competitive Employment of Autistic Adults Through Late Midlife. The manuscript addresses a pressing issue, competitive employment, within the autistic community and provides valuable data that can assist future researchers, policymakers, and families/individuals in understanding employment patterns in autism through midlife and planning for adult lives.  Below are a few comments/suggestions for author consideration.

  1. Table 2 showed a gradual increase in %age of full sample engaged in competitive employment through time 5 followed by a decrease in %age at time 6.  Were the authors able to view data and determine possible reasons for this dip or was it random? The dip was in both ID and No ID.  

Response: We thank this reviewer for this observation. We believe that the change between Time 5 and Time 6 is a brief deviation from the overall trend. We considered whether Time 5 to Time 6 straddled the Great Recession, on which some of our previous research has focused (Hickey, in press). However, those time points were after the Great Recession. This “blip” may have reflected unknown changes in policies or the political climate, or could be due to socio-cultural factors that we cannot determine with our available data.

It is important to note that, although data are presented by time point in Tables 1-3 for the readers’ information, the main analyses (i.e., using accelerated longitudinal design) are based on age and not time point of data collection. The goal of the analyses was to measure individual change as people age. We did not find any cohort effects, suggesting that the brief decrease in the percentage of the full sample engaged in competitive employment at Time 6 should not affect the trajectory analysis, as the participants who contributed data were at various ages at Time 5 and Time 6.

Hickey, E. J., Smith DaWalt, L., Bolt, D., Hong, J., Song, J., Lounds Taylor, J., & Mailick, M. R. (in press). The impact of the great recession on autistic adults and their mothers. American Journal on Intellectual and Developmental Disability.

  1. Based on the data outcomes, particularly for autistic individuals with ID, do the authors have recommendations/thoughts on the practical implications for the population as well as schools/vocational rehab and other agencies in their preparation programs.  

Response: Thank you for this important question. Our findings offer autistic individuals and their family members, as well as schools, vocational rehabilitation workers, and other agencies, data from which to establish expectations for the reality of what competitive work may look like for autistic adults. Our results do not show a typical career progression. Rather, for those without ID, the number of competitive employment hours increased from young adulthood until early midlife, then leveled off and decreased into late midlife. For those with ID, engagement in competitive employment was low throughout adulthood.  Further, even among those who have worked competitively, that work is often inconsistent, as indicated by the frequent number of “zero” hours across waves (see Figure 2). Many factors undoubtedly contribute to these patterns, including the preferences and abilities of the autistic adult as well as labor market conditions. If having a stable competitive job is the goal for an autistic adult, it may not be possible for the position to be full time. We now address these points in the Conclusion section on page 15.

  1. It would be interesting to see how many of the individuals in your study had obtained integrated community jobs during their transition years in school (age 14-21) and whether having a community job while in high school predicted increased competitive employment engagement.
  2. It appeared that regardless of ID status, autistic individuals are underemployed in terms of hours.   The assumption would be that the pay for the sample would not be enough to sustain them in independent living situations.  This may be a viable recommendation for future research in this area along with ideas for intervention research that can change the trajectories.

Response: We thank this reviewer for this thought-provoking comment. We agree with the reviewer that, based on these findings, it is unlikely that autistic individuals such as those in this sample would be able to sustain independent living with pay from competitive work. Our data suggest that competitive employment may not be the pathway to financial independence for many autistic adults, even among those without intellectual disability. In future studies, it will be important to collect data about the preferences of autistic adults for full time employment. Working full-time might not be feasible or desirable for everyone. Policies that provide comprehensive financial support that realistically consider the level of employment that individuals and their families prefer or are able to obtain should be considered.  We address this in the Discussion and Conclusion sections on pages 14 and 15.

Reviewer 2 Report

Comments and Suggestions for Authors

This is an interesting study and a well-written paper.  I have very few comments or concerns but these are below: 

1. A third party (the adult's mother) reported the number of hours worked.  Do the researchers know of any studies where the accuracy of such reports was evaluated?  If so, that would be worth including. 

2. It would be interesting to know the authors' thoughts as to the potential importance of this study to a working job coach or other DSP.   Do the outcomes of this study have the potential to impact, in a positive way, the work of job coaches and competitive employees? 

Author Response

Reviewer 2

This is an interesting study and a well-written paper.  I have very few comments or concerns but these are below: 

  1. A third party (the adult's mother) reported the number of hours worked.  Do the researchers know of any studies where the accuracy of such reports was evaluated?  If so, that would be worth including. 

Russell, G., Mandy, W., Elliott, D., White, R., Pittwood, T., & Ford, T. (2019). Selection bias on intellectual ability in autism research: A cross-sectional review and meta-analysis. Molecular Autism10(1), 1-10. doi: 10.1186/s13229-019-0260-x

  1. It would be interesting to know the authors' thoughts as to the potential importance of this study to a working job coach or other DSP.   Do the outcomes of this study have the potential to impact, in a positive way, the work of job coaches and competitive employees? 

Response: Although this analysis did not include supported employment, our findings offer autistic individuals and their family members, as well as schools, vocational rehabilitation workers, and other agencies, data from which to establish expectations for the reality of what work may look like for autistic adults. This may be informative for job coaches or other DSPs within supported work systems as well. Our results do not show a typical career progression of competitive work. Rather, for those without ID, the number of competitive employment hours increased from young adulthood until early midlife, then leveled off and decreased into late midlife. For those with ID, engagement in competitive employment was low throughout adulthood.  Further, even among those who have worked competitively, that work is often inconsistent, as indicated by the frequent number of “zero” hours across waves (see Figure 2). Many factors undoubtedly contribute to these patterns, including the preferences and abilities of the autistic adult as well as labor market conditions. If having a stable competitive job is the goal for an autistic adult, it may not be possible for the position to be full time. We now address these points in the Conclusion section on page 15.

Reviewer 3 Report

Comments and Suggestions for Authors

This is a well-written paper. I have few minor comments for the improvement of this paper:

1. Please check reference style cited in the text. It does not follow MDPI guidelines.

2. The discussion needs further elaboration on the implications for practice.

3. At the current stage, discussion is rather short. It would be good to systematically report the discussion by first highlighting the core summary findings, discussion about consistencies with previous works, plausibility based on available frameworks, strengths and limitations, and conclusions.

Comments on the Quality of English Language

Acceptable

Author Response

Reviewer 4

This is a well-written paper. I have few minor comments for the improvement of this paper:

  1. Please check reference style cited in the text. It does not follow MDPI guidelines.

Response: We have edited the in-text citations throughout the manuscript.

  1. The discussion needs further elaboration on the implications for practice.

Response: Thank you for this point. We have edited the discussion and added several examples of implications for practice, specifically for autistic adults and their families, as well as schools/vocational rehabilitation, and other agencies, in their preparation programs.

  1. At the current stage, discussion is rather short. It would be good to systematically report the discussion by first highlighting the core summary findings, discussion about consistencies with previous works, plausibility based on available frameworks, strengths and limitations, and conclusions.

Response: We thank the reviewer for this feedback and have thoroughly revised the discussion to add more content, as well as rearrange, as suggested. See, in particular, pages 14 and 15.